# Impact of Green Servant Leadership in Pakistani Small and Medium Enterprises: Bridging Pro-Environmental Behaviour through Environmental Passion and Climate for Green Creativity

Syed Haider Ali Shah [1], Mochammad Fahlevi [2,*], Eman Zameer Rahman [3], Muhammad Akram [1], Kamran Jamshed [3], Mohammed Aljuaid [4] and Jaffar Abbas [5]

1   Business Studies Department, Bahria University, Islamabad 44000, Pakistan; haidershah11@gmail.com (S.H.A.S.); makram@bahria.edu.pk (M.A.)
2   Management Department, BINUS Online Learning, Bina Nusantara University, Jakarta 11480, Indonesia
3   Department of Management Sciences, Bahria University, Islamabad 44000, Pakistan; emansadiya@gmail.com (E.Z.R.); kamran.jamshed32@gmail.com (K.J.)
4   Department of Health Administration, College of Business Administration, King Saud University, Riyadh 11437, Saudi Arabia; maljuaid@ksu.edu.sa
5   School of Media and Communication, Shanghai Jiao Tong University (SJTU), Shanghai 200240, China; dr.j.abbas@outlook.com
*   Correspondence: mochammad.fahlevi@binus.ac.id

**Abstract:** Small and medium enterprises (SMEs) are crucial for any economy to grow and succeed, as evidenced in the Pakistani context, where SMEs contribute 30% of the country's GDP. The objective of this study is to link green servant leadership (GSL) to the pro-environmental behaviour (PEB) of employees, particularly in SMEs, which very few studies have investigated. Building on social learning theory (SLT), this study developed and tested the conceptual framework that examines the impact of GSL on PEB and the mediating role of environmental passion (EP) and climate for green creativity (CFGC) among employees of SMEs. Data were collected from 460 middle-line managers, and a structural equation modelling (SEM) technique was applied to test hypotheses. The main findings revealed that GSL is impacted by the PEB, while EP and CFGC mediated these relations. The study findings demonstrated that a GSL with strong practices and values towards the environment could have a significant impact on employees' PEB. This study fills the research gaps in different ways: First, by identifying the role of GSL on the PEB of employees. Second, by examining the dual-mediation mechanism of EP and CFGC between GSL and PEB. Third, this study is focused on the economic context of a developing country. The study offers guidelines for establishing PEB in SMEs. Multiple training programmes and effective planning procedures can achieve this milestone. The administration of SMEs should also give special consideration to pro-environmental issues in its hiring and recruitment practices for managers and leaders.

**Keywords:** green servant leadership; pro-environmental behaviour; environmental passion; climate for green creativity; SMEs; Pakistan



## 1. Introduction

Nowadays, organizations are taking multiple steps to prevent harmful practices towards the natural environment and taking initiatives to promote environmentally friendly practices, as it has become the centre of attention for the organization to consider sustainability and pro-environmental behaviour (PEB) [1–3]. This type of employee conduct refers to voluntary actions that contribute to the environmental sustainability of the employer's organization [4,5].

PEB refers to the practices that are helpful in promoting the natural environment through different practices of recycling, reusable initiatives, reprocessing, rebuilding, and the applications of different ideas to implement practices that reduce the harmful effect of an organization towards the environment by adopting the practices of green products and processes [1,6,7]. Organizations that are involved in PEB practices are in a better position to gain a competitive edge over their competitors, as pro-environmental practices reduce costs, generate revenue, and help create a positive image through certain practices towards sustainability [8,9]. Therefore, there is much curiosity among researchers to investigate further the antecedents and underlying mechanisms that promote PEB [1,10,11].

The role of leaders is immensely important in shaping employees' behaviour [12–15]. However, there is a lack of understanding and empirical evidence to uncover the underlying mechanisms of leadership roles on enhancing PEB [12,16,17], particularly green servant leadership (GSL) [1,2], which has been reported as a significant predictor of PEB [18,19]. GSL places a higher emphasis on environmental advantages, both for the leader personally and the organization, while simultaneously focusing on instilling pro-environmental values in key organizational stakeholders, such as employees and customers [3,7,12]. Various studies have been conducted on other styles of leadership like green transformational leadership [20], ethical leadership [21], and responsible leadership [6]. Therefore, there is a gap in the empirical evidence to investigate GSL's impact on PEB [2,22,23]. The novelty of the study lies in bridging the gap in the literature by focusing on the environmental aspect of GSL that inculcates and promotes the PEB.

The role of passion is significant in carrying out different activities with dedication. Employees exhibit passion in multiple ways demonstrating positive emotions at work, developing meaningful connections to different work tasks, and being intrinsically directed to accomplish the tasks [24]. Environmental passion (EP) is the employee's emotional experiences towards the various environmental practices in the organization [20]. A strong passion for the environment serves as a motivating force that drives individuals to engage in pro-environmental behaviour [25]. Employees who are passionate about environmental causes not only engage in spontaneous pro-environmental actions but also maintain a consistent commitment to PEB, and exhibit themselves as environmentalists [26]. Employees with strong EP are not only inclined towards the PEB but also identify themselves as environmentalists [20,26]. The role of the leader is pertinent to further strengthen the EP in employees, which in turn can promote PEB. GSL specifically prioritizes environmental concerns and acknowledges employees' contributions to the community, particularly through environmentally conscious actions [27], which can enhance their EP as conscientious environmental citizens and portray PEB in the organization [12]. There is a lack of studies and gaps that investigate the mediation mechanism of EP between GSL and PEB, particularly in the SMEs of Pakistan.

To promote PEB, the role of climate for green creativity (CFGC) is also crucial [28,29]. CFGC refers to organizational support to employees to achieve their outcomes through their own creative manner [28,30,31]. Particularly, when employees believe that they are appreciated and rewarded at a workplace for their creativity, such practices enhance the notion of the climate for creativity [22,32]. GSL is regarded as an architect of environments that promote a climate conducive to creativity [33]. Moreover, it provides essential resources and offers support for creative and innovative endeavours [28]. Servant leadership was found to have positive relationship with group creativity [34]. In an environment where there is an abundance of resources, comprehensive support, and attractive incentives for novel ideas, employees are more inclined to engage in innovative behaviours while considering the environment [28]. However, there is a limitation in the literature regarding the mechanism underlying the mediation of CFGC between GSL and PEB [28,35].

This study has several contributions and objectives of this study: First, this is to investigate GSL's impact on PEB, which is rarely investigated in the context of Pakistan, which is a developing country. Second, drawing on the theory of social learning theory, this study examines the underlying mechanism of EP between GSL and PEB and integrates

the holistic research work in one framework, which is another contribution of this study. The third contribution of the study lies in the investigation of the mediating mechanism of the climate for green creativity between GSL and PEB in the Pakistan context.

## 2. Theoretical Background and Hypotheses Development

### 2.1. Pro-Environmental Behaviour

PEBs are generally regarded as to be human behaviours that are developed and used sustainably in relation to the environment or that attempt to lessen the negative effects of those behaviours on the environment [20,36], as well as the performance of actions that are good for the environment and the avoidance of actions that are bad for the environment [37]. Some studies have referred to PEB as the environmentally friendly behaviours of employees at the workplace, such as deliberately recycling paper, conserving water and electricity, etc., which are relevant to the organizational management setting [15]. Further, PEB is defined as activity that modifies the ecosystem's structure and dynamics and has a positive impact on the availability of resources like energy or materials [38].

Employees' PEB is crucial to achieving an organization's sustainability goals and halting the degradation of the environment [2]. Additionally, it is posited that the conduct observed within the organization, which deviates from formal policies and prescribed implementation procedures, is undertaken voluntarily by employees [4]. In a nutshell, employees have discretion and complete freedom to exhibit behaviour that is proactive toward environmental preservation at the individual level. Businesses have started to recognize the connection between environmental protection, as well as their persistence and performance [39].

Cultivating PEB can benefit firms strategically by increasing efficiency, increasing income, improving brand perception, achieving sustainability goals, and giving them an edge over its competitors [2]. Nonetheless, both academics and practitioners have neglected the topic of employee PEB. Employees' pro-environmental conduct is defined as their desire to participate in environmental engagement [40]. One of the major predictors of employee PEB is the leadership style in a business, according to past research studies [20,26,41].

### 2.2. Green Servant Leadership and Pro-Environmental Behaviour

The focus and the centre of attention for servant leadership is the employees' interest [1,42], directing them in a way to pursue their needs [43], and carrying out different activities that exhibit him/her as a role model of compassionate, altruistic love and empathy [1]. Servant leadership has received much attention from scholars regarding its impact on various aspects of organizational, group, and individual outcomes [1].

Very few studies have focused the environmental concerns and issues and their relation to servant leadership [2,35]. Luu Trong Tuan's [31] idea of integrating the environmental component with servant leadership in the literature says that PEB should serve as an example of humility, sincerity, interpersonal acceptance, and stewardship toward employees' pro-environmental accomplishments. It should also give advice, help people become pro-environmental citizens, and encourage this kind of behaviour in others. This reflects the pro-environmental approach of the leaders, which can be termed as green leadership. Thus, this study takes this concept as GSL [1,35].

In addition to that, the various concepts of green practices, particularly green human resource management (GHRM) practices, are also part of the organizational initiatives, and to implement green practices properly, the role of leaders is critical, and plays an integral part in impacting the organization and employees [5]. Furthermore, Robertson and Barling [44] advocated that servant leadership is significant in promoting PEB, and after the article appeared, most of the researchers turned to investigating the servant leadership in the context to the environment, which is now referred to as GSL [1,12,35].

PEB is reflected in the positive initiatives of the employees with respect to energy, material, nature, and ecosystem [38,45]. Further, it is discussed in the literature as behaviour that is reflected in terms of reuse, recycling, and conservation of the energy [46].

Employees who are directed for PEB benefit the organizational performance directly and indirectly [1,2]. In this study, the concepts of GSL and PEB are taken and linked from the perspective of SLT [47]. The green philosophy paves the way for the implementation of such practices, which opens the venue to cultivate and develop green values in employees through leadership [35].

Based on above discussion, the following hypothesis has been developed:

**Hypothesis 1 (H1).** *GSL positively relates to PEB.*

### 2.3. Mediating Role of Environmental Passion

In the literature review, EP is investigated and linked with environmental issues [28]. EP is considered as the motivational state related to the environment. Passion is referred to as the state where employees dedicate themselves in a way to pursue certain tasks and activities persistently [48]. Studies have advocated that such a source of pursuing dedication be utilized by the organization to maximize the organizational performance [49]. When such emotions are directed towards environmentally friendly practices, they develop a state of EP in which they care about the environment and redirect their behaviour towards environmental conservation practices [25,50].

GLS attracts employees to work voluntarily, and is friendly towards the environment [1]. The purpose of GSL is to inculcate the emotions in employees towards environmentally friendly practices [51]. Furthermore, it follows the directions of the philosophical perspective from SLT that followers tend to follow leaders and reciprocate through their behaviour and actions. Based on the above argument, it can be referred to as GSL leading employees towards environmentally friendly practices [31]. Furthermore, transformational leadership behaviours, whether verbal or nonverbal, emphasizing environmental issues can be viewed as affective events that are essential for igniting the EP of subordinates [15,20].

Additionally, by displaying protection of the environment, environmentally focused transformational leaders can show employees the organization's commitment and conviction, potentially raising their positive emotional anticipation of environmental operations [52]. In addition to that, environmentally focused transformational leaders who exhibit high levels of idealized influence and intellectual stimulation encourage people to find creative solutions to environmental problems and direct staff to put organizational social responsibility and environmental sustainability ahead of their own self-interests, increasing staff's intrinsic motivation to participate in environmental protection activities [20,53].

Environmentally focused transformational leaders who display interactive processes (such as compassion and mentoring) for their employees to be more receptive to their leadership's guidance on environmental issues and to participate in environmental protection activities should also make them feel effective and practical support [54]. Employees are much less likely to demonstrate a passion for environmental issues if they have limited opportunities to observe or support environmentally beneficial actions exhibited by their leaders [20,29]. A study found a relationship between transformational leadership and EP [15]. Similarly, other studies also found that other types of leadership impact EP [25,50,55]. GSL promotes employees' behaviour to serve the organization as well as the community through environmental practices [56]. The aim of GSL is to develop the emotional state in which employees always want to protect the environment while using fewer resources and less energy and adopting recycling practices [35]. GSL exhibits PEB, which in turn influences the employees in a way to develop behaviour that cares for the environment [2,25,35]. Most studies investigated the mediating role of EP between other styles of leadership on PEB [15,20]. Based on the above discussion, it is important to investigate the mediating role of EP between GSL and PEB, and the following hypotheses have been developed.

**Hypothesis 2 (H2).** *GSL positively relates to EP.*

**Hypothesis 4 (H4).** *EP positively relates to PEB.*

**Hypothesis 6 (H6).** *EP mediates the relationship between GSL and PEB.*

*2.4. Mediating Role of Climate for Green Creativity*

CFGC is referred to as organizational support to employees to carry out their activities in a way in which they work in a creative manner [28]. In this study, CFGC reflects on an environment where employees are treated and rewarded based on their creative manner to use fewer resources and new initiatives to solve problems and adopt environmentally friendly behaviour [32]. Moreover, CFGC is considering the utilization of organizational resources and providing support for green creativity, and is involved constantly in developing new insights regarding green change, green practices, green initiatives, and reward systems to employees accordingly for their green creative work-related outcomes [25]. Moreover, the allocation of financial resources by the organization for green projects and initiatives is referred to as CFGC. It has been argued that the values of a green environment convey to employees the need for them to act sustainably [57]. By giving employees the social resources, they require a setting that encourages innovation in the green movement, values, and pro-environmental perception to be merged, which may motivate employees to channel their pro-environmental resources into green work-related behaviours [28].

The role of GSL is very important for cultivating a green environment for employees. Employees have a greater tendency to positively perceive CFGC when environmentally friendly leaders invest in creating it through the provision of adequate green resources, and as a result, they are more likely to display PEB [28]. There are various reasons why this study has focused on GSL as leadership style. First, GSL is regarded as the architecture of the organization that cultivates and supports CFGC [33]. In addition to that, they encourage and reward the employees for green creativity [31,58], and give them the opportunity to solve problems in their own creative manner, and embed this in the culture of the organization [59]. Another reason is that GSL develops the organizational environment in a way in which employees feel free to develop and share their ideas and perform experiments regardless of the success or failure of the idea because they feel support from GSL for green creativity [58,60].

Although some efforts have been made to investigate the role of a green climate as a mediating factor between environmentally conscious servant leadership and green performance [22], the role of a green climate as a mediating factor for green creativity in the hospitality sector has not yet been investigated [28,30]. Moreover, in general, the literature on green workplaces argues that employees' views of a workplace's green environment might influence their attitudes and behaviours [61,62]. Another reason is that by giving discretion to employees to utilize resources in a way in which employees feel empowered, and ultimately develop feelings that employees are given resources and empowerment to transfer resources the way they want for green creativity, such decisions reflect CFGC. SLT also advocates the same: that employees tend to follow their leaders' actions, and environmentally friendly practices performed by leaders are like catalysts to boost their confidence level and lead them to exhibit support and care for organizational CFGC. However, there is a lack of empirical studies on CFGC mediating the relationship between GSL and PEB. Hence, this study proposes the following hypotheses:

**Hypothesis 3 (H3).** *GSL positively relates to CFGC.*

**Hypothesis 5 (H5).** *CFGC positively relates to PEB.*

**Hypothesis 7 (H7).** *CFGC mediates the relationship between GSL and PEB.*

The research model is presented as follows (see Figure 1).

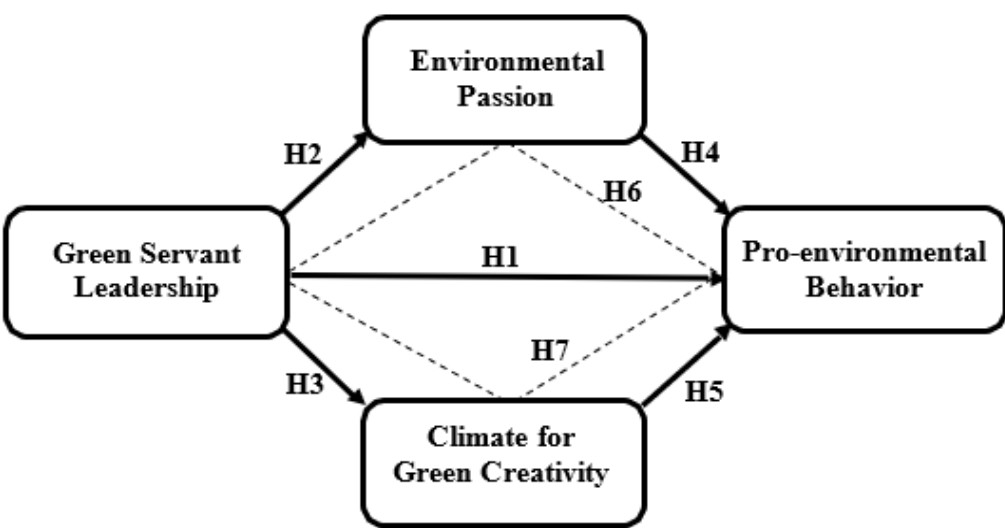

**Figure 1.** Conceptual framework.

### 3. Method

*3.1. Sample and Methodological Procedures*

Middle line manager is the unit of analysis working in SMEs. Most of the SMEs are operating in the major city and province of Pakistan, which is Punjab, representing approximately 66% of SMEs. According to SMEDA [63], In Punjab, there were more than 15,000 SMEs that were registered. There are four categories: Wood and Furniture, Sports, Leather and Footwear, and Textiles, with a total of 430 businesses. A cluster sampling technique was used to gather information from 460 middle-level managers [16,64]. Based on their share of the overall population, the four industries indicated above were clustered together to create 850 surveys. The distribution and collection of questionnaires are shown in Tables 1 and 2.

**Table 1.** Industry profile.

| Sr. No. | Industry | % | Firms |
|---------|----------|---|-------|
| 1 | Textile | 16.60 | 80 |
| 2 | Leather/Footwear | 16.00 | 80 |
| 3 | Sports | 15.70 | 75 |
| 4 | Wood and Furniture | 16.51 | 72 |
| Total | | 100 | 430 |

**Table 2.** Distributing and collecting questionnaires.

| Cluster | Questionnaires Distributed | Questionnaires Received |
|---------|---------------------------|-------------------------|
| Lahore | 180 | 120 |
| Multan | 160 | 84 |
| Sialkot | 150 | 74 |
| Gujrat | 140 | 70 |
| Faisalabad | 120 | 70 |
| Gujranwala | 100 | 52 |
| Total | 850 | 470 |

*3.2. Data Collection Instruments*

To assess GSL measured through a twelve-item scale adopted from a study by Liden, Wayne, Zhao, and Henderson [43], the sample statements are "My manager emphasizes the importance of contributing to the environmental improvement" and "I am encouraged by my manager to volunteer in the environmental activities". Items for the PEB scale were taken from Robertson and Barling [15]. There were a total of 12 items used to assess

PEB. An example is "At work, I take stairs instead of elevators to save energy". Moreover, 10-item scale established by Roberson and Barling [15] was used to assess EP. Sample items include "I am passionate about the environment". In order to measure CFGC, this study adopted the 5-item scale from Kim and Yoon [32]. The sample statement is "I am passionate about the environment".

## 4. Results

Before proceeding with the main analysis of the study, it is crucial to understand the demographics of the participants as represented in Table 3. The demographics table provides a clear breakdown of the respondents in terms of gender, age, education, and experience. Data normality is the first condition before proceeding towards the main analysis. This research study first detects the missing values, and an imputation approach was employed for mean substitution [65]. For a further data normality check, this study also used the indices of kurtosis and skewness measures, and they were 2.201 and 1.374, respectively, with data found to be in normal distribution [66]. Another important aspect of the analysis is to check the common method biases, and this study checks it through the Harman single factor. According to this analysis, the characteristic root of the common factor with the highest explanatory power is 11.582, which represents 40.541 of the overall variation in the absence of factor rotation. It portrays no common method bias because not a single factor explained most of the dependent and independent variables. Table 4 represents the descriptive statistics along with correlation and standard deviation values.

**Table 3.** Demographics of the study.

| Demographics | No. of Respondents | Percentage (%) |
|---|---|---|
| **Gender:** | | |
| Male | 326 | 71 |
| Female | 134 | 29 |
| **Age:** | | |
| Less than 40 years | 312 | 68 |
| More than 40 years | 148 | 32 |
| **Education:** | | |
| Bachelors | 386 | 84 |
| Masters. | 74 | 16 |
| **Experience:** | | |
| <5 years | 165 | 36 |
| 5–10 years | 203 | 43 |
| >10 years | 97 | 21 |

**Table 4.** Correlations.

| No | All Variables | Mean (SD) | 1 | 2 | 3 | 4 |
|---|---|---|---|---|---|---|
| 1 | Green Servant Leadership (GSL) | 3.72 (0.81) | **(0.890)** | | | |
| 2 | Climate For Green Creativity | 3.33 (0.77) | 0.242 * | **(0.870)** | | |
| 3 | Environmental Passion | 3.73 (0.98) | 0.559 ** | 0.131 * | **(0.901)** | |
| 4 | Pro-Environmental Behaviour (PEB) | 3.49 (0.78) | 0.651 * | 0.440 ** | 0.565 * | **(0.820)** |

** Correlation is significant at the 0.01 level (2-tailed). * Correlation is significant at the 0.05 level (2-tailed). Values in bold are the Cronbach alphas. SD, standard deviation.

### 4.1. Measurement Model

To ensure the validity (convergent and discriminant validity), the CFA series were employed (see Table 5). SPSS AMOS 24 was employed to check the model fitness for all variables through structural equation modelling (SEM) [67]. The four-factor model (GSL, EP, CFGC, and PEB) was found to be a superior fit to the data among all the three-factor, two-factor, and one-factor models, as all components were loaded on a single factor (see Table 4). Factor loading has met the minimum requirements in the new model, namely at

least more than 0.5 for each item in the construct (see Table 6) [65]. Along with that, the Cronbach's alpha values were also found to be above 0.70 (see Table 6). In addition to that, AVE values that are greater than 0.50 also show the convergent validity of the model [68]. Table 7 shows the discriminant validity, which is within the range of acceptability [68]. Another important aspect of data analysis is to check the multicollinearity issue; in this study, there was no multicollinearity issue, as the values of VIF are within the range of 1.89 to 4.03 (less than 10).

**Table 5.** Results of model comparisons using a CFA approach.

| Model | $\lambda2$ | df | TLI | CFI | IFI | NFI | RMSEA | SRMR |
|---|---|---|---|---|---|---|---|---|
| (MO) | 584.414 | 234 | 0.954 | 0.955 | 0.959 | 0.884 | 0.053 | 0.0491 |
| (M1) | 110.586 | 57 | 0.910 | 0.901 | 0.941 | 0.965 | 0.065 | 0.0390 |
| (M2) | 55.257 | 41 | 0.941 | 0.950 | 0.996 | 0.935 | 0.058 | 0.0384 |
| (M3) | 70.451 | 10 | 0.847 | 0.851 | 0.854 | 0.865 | 0.228 | 0.0558 |

**Table 6.** Construct validity.

| Construct | Items | Factor Loading | AVE | CR | Cronbach's Alpha |
|---|---|---|---|---|---|
| GSL | 12 | 0.59–0.89 | 0.55 | 0.87 | 0.89 |
| EP | 10 | 0.63–0.84 | 0.52 | 0.82 | 0.90 |
| CFGC | 5 | 0.65–0.87 | 0.57 | 0.62 | 0.87 |
| PEB | 12 | 0.55–0.87 | 0.52 | 0.81 | 0.82 |

**Table 7.** Discriminant validity.

| Constructs | CR | AVE | MSV | MaxR(H) | GSL | PEB | CFGC | EP |
|---|---|---|---|---|---|---|---|---|
| GSL | 0.874 | 0.552 | 0.382 | 0.928 | 0.701 | | | |
| PEB | 0.812 | 0.527 | 0.474 | 0.904 | 0.554 | 0.679 | | |
| CFGC | 0.623 | 0.574 | 0.489 | 0.879 | 0.638 | 0.702 | 0.810 | |
| EP | 0.824 | 0.528 | 0.481 | 0.902 | 0.547 | 0.679 | 0.723 | 0.795 |

Green servant leadership (GSL); environmental passion (EP); climate for green creativity (CFGC); pro-environmental behaviour (PEB); AVE = average variance extracted; MSV = maximum shared variance; MaxR(H) = McDonald construct reliability.

### 4.2. Structural Model

The model was evaluated and goodness-of-fit measures were achieved [69], where $\chi^2 = 269.654$, *df* = 100, $\chi^2/df = 1.347$, [RMSEA] = 0.053, [GFI] = 0.958, [AGFI] = 0.954, [NFI] = 0.914, [RFI] = 0.935, [IFI] = 0.965, [TLI] = 0.952, [CFI] = 0.935. In this structural model, it is known that all direct (see Table 8) and indirect (see Table 9) influences have a positive and significant effect, so these results can conclude that the entire hypothesis in this study is accepted. Specifically, the perceived direct effect of GSL on PEB has the highest standard path loading, and the mediating role of EP has the highest standard path loading for the indirect effect. These results explain the magnitude of the role of GSL on PEB and the success of EP in mediating the effect of GSL on PEB in this research model.

**Table 8.** Path analysis.

| Hypothesis | Predicted Relationship | Standard Path Loadings | Standard Error | t-Value | p-Value | Decision |
|---|---|---|---|---|---|---|
| H1 | Perceived GSL → PEB | 0.68 | 0.08 | 5.912 | 0.003 | Supported |
| H2 | Perceived GSL → EP | 0.50 | 0.10 | 6.012 | 0.005 | Supported |
| H3 | Perceived GSL → CFGC | 0.61 | 0.09 | 6.587 | 0.001 | Supported |
| H4 | EP → PEB | 0.58 | 0.82 | 5.258 | 0.003 | Supported |
| H5 | CFGC→ PEB | 0.57 | 0.25 | 6.812 | 0.003 | Supported |

Green servant leadership (GSL); environmental passion (EP); climate for green creativity (CFGC); pro-environmental behaviour (PEB).

**Table 9.** Mediation effects.

| Hypothesis | Parameter | Estimate | Lower Bound | Upper Bound | p-Value | Decision |
|---|---|---|---|---|---|---|
| H6 | Panel Ia Perceived GSL → EP → PEB | 0.541 | 0.255 | 0.439 | 0.001 | Supported |
| H7 | Panel IIb Perceived GSL → CFGC→ PEB | 0.483 | 0.174 | 0.374 | 0.003 | Supported |

Green servant leadership (GSL); environmental passion (EP); climate for green creativity (CFGC); pro-environmental behaviour (PEB). (a) goodness of fit: $\chi^2/df$ = 1.960, RMSEA = 0.052, GFI = 0.957, CFI = 0.958. ("GSL → EP" was constrained to be zero). (b) Goodness of fit: $\chi^2/df$ = 2.445, RMSEA = 0.051, GFI = 0.941, CFI = 0.959. ("GSL → CFGC→ PEB" was constrained to be zero).

## 5. Discussion and Conclusions

This study advances the discussion on PEB, especially in the context of energy, by considering Pakistan's environmental fragility and the role that citizens play in maintaining the environment and ecosystem. The findings demonstrated that GSL can have an impact on employees' PEB inside a firm.

Different organizational components, such as leadership style and a welcoming green workplace, might influence the energy consumption behaviour of SME workers in this industry [16]. This means that leaders who put sustainability first and encourage and help their employees to perform things that are good for the environment can have a big impact on how employees in the SME sector act. The fact that SMEs use a lot of energy to make things and provide services suggests that these businesses may have a big effect on the environment [70,71]. Because of this, it is important to think about how organizational factors like leadership style and a friendly, green workplace affect the way SME workers in this industry use energy. A leadership style that emphasizes sustainability and servant leadership can encourage employees to perform things that are good for the environment, like using less energy, recycling, and making less trash [72].

This study's findings demonstrated that GSL with strong practices and values towards the environment could have a significant impact on employees' PEB ($\beta$ = 0.687), supporting H1. GSL can be defined as leadership that values environmental advantages over financial ones. It also demonstrates its dedication to the environment by its energy consumption tendencies. Through the lens of SLT, employees are involved in energy conservation activities when they observe that their leader cares and promotes PEB, which ultimately motivates them to exhibit PEB into practice. The findings indicated that GSL positively impacts on EP and CFGC, supporting H2 and H3. This suggests that GSL incites EP and

CFGC, which further encourages individuals to develop favourable perceptions of the environment and exhibit PEB [28]. Another interesting finding is that CFGC was found to have a positive and favourable impact on PEB, supporting H4, and EP was found to have a positive impact on PEB, supporting H4 and H5, respectively. The findings are consistent with the SLT, which holds that followers mimic leaders by adopting their behaviour and actions, and that this leads to employees adopting green behaviour, which prompts them to participate in PEB [28]. The statistical findings in this regard demonstrated that EP not only significantly mediates the association between GSL and PEB ($\beta$ = 0.541), but also directly affects PEB (b = 0.58), supporting H6. Similarly, the findings show the mediation effect of CFGC ($\beta$ = 0.483), between GSL and PEB ($\beta$ = 0.483), and directly affecting PEB ($\beta$ = 0.57), supporting H7. This implies that the influence of GSL on fostering PEB is intricate, with a mediating mechanism playing a pivotal role in this process [73]. This led to the notion that GSL not only guides and facilitates their employees to reduce their energy consumption but also enhances employees' perceptions of how their energy use can help their firm to have an impact on their PEB. GSL emerges as a valuable leadership approach, as it nurtures environmental enthusiasm among employees, subsequently leading them to exhibit PEB [25]. GSL incites EP about the environment, and as a result, such employees exhibit PEB [20,25].

EP refers to employees' emotional attachment and commitment to environmental sustainability. "CFGC" refers to the organizational culture that encourages and supports innovation and creativity in developing environmentally sustainable practices. These factors can mediate the effect of GSL on PEB by influencing the motivation and ability of employees to engage in environmentally sustainable practices. GSL may have a significant impact on PEB, and this relationship may be facilitated by factors such as EP and CFGC. It is evident that a green creativity-friendly environment serves as a mechanism for conveying expectations to employees regarding PEB, such as organizational citizenship behaviour for the environment [74,75]. By fostering a culture of environmental sustainability and providing employees with the necessary resources and support, SMEs in Pakistan can promote PEB among their employees, which can have significant environmental and social benefits. The purpose of the proposed model was to investigate the effects of GSL on PEB directly and through the mediating role of EP and CFGC. The findings of the study are consistent with the previous studies [2,28,33]. GSL offers employees abundant and valuable green resources, consequently stimulating their perception of a climate conducive to creativity. This, in turn, motivates employees to participate in PEB [76]. Previous studies have focused on organizational practices to promote PEB, but this study focused on the role of GSL on overall PEB.

EP is important for SMEs in Pakistan because it can be a strong motivator for employees to act in ways that are good for the environment. When employees have a strong emotional connection to and commitment to protecting the environment, they are more likely to take the initiative and go above and beyond what is expected of them at work to perform things that are good for the environment. When it comes to SMEs in Pakistan, where there may be a lack of resources and skills for putting environmentally sustainable practices into place, passion for the environment can be a big driver of change. By encouraging employees to care about the environment, SMEs can create a culture where sustainability is valued and put first. This can make people more motivated and committed to protecting the environment, which can lead to less energy use, less waste, and other actions that are good for the environment. Also, in a country like Pakistan, where environmental problems like air and water pollution, deforestation, and climate change are big, promoting EP can help raise awareness and motivate employees, customers, and the wider community to take action. In the end, encouraging SME employees in Pakistan to care about the environment can help build a more sustainable and resilient future.

Furthermore, the CFGC is important for SMEs in Pakistan because it can foster an organizational culture that encourages and supports innovation and creativity in developing environmentally sustainable practices. Promoting CFGC can also help SMEs in Pakistan

overcome barriers such as a lack of knowledge, experience, and resources. By encouraging employees to develop and test new ideas and solutions, SMEs can tap into the collective knowledge and creativity of their workforce to find practical and effective ways to reduce their environmental impact.

*5.1. Theoretical Implications*

This study added several new insights to the body of knowledge about environmental management in SME settings. First, this study filled the critical knowledge gap and addressed the scholarly call made by Al-Ghazali et al. [16] to further explore how GSL may encourage PEB in the SEM sector. Findings of this study provided empirical evidence that GSL is capable of effecting the desired change. This study adds to the existing body of knowledge. Furthermore, employees are positively influenced by the environmental concern displayed by their leader through GSL [77]. Second, there is a lot of attention being paid to the research on the antecedents of PEB, which is still considered as it is in the early stages and needs further investigations [3,7,16]. Therefore, this study highlights the antecedents of PEB, namely EP and CFGC, for PEB in SMEs.

Moreover, the study revealed that GSL instigates EP, which, in turn, motivates employees to adopt PEB. The leader's genuine environmental care influences the followers' passion for the environment [20,78–80]. Third, although research in SMEs has shown that GSL is a key factor in employees' pro-environmental activities, little is known about the underlying mechanisms that control these associations [45]. Furthermore, it was found that the GSL is a leadership approach that can effectuate such desired transformation. Employees are affected by the environmental awareness demonstrated by their leader through GSL [12]. Fourth, this study adds to the SLT in SMEs because there is little evidence regarding how employees' EP and CFGC lead to PEB and their conceptualization. As servant leaders provide green-related resources, these resources are disseminated among employees, ultimately fostering positive perceptions among them regarding the environment's creative climate. When environmentally focused GSL offers ample environmental resources, it can act as a catalyst for resource exchange among employees. This, in turn, contributes to the development of a shared mental model and ultimately enhances a positive perception of the organizational climate [81]. Fifth, no current study has attempted into the influence of EP in mediating the relationship between GSL and PEB in the context of SMEs, particularly [1,20,25,29,30]. According to the findings of this study, SMEs may significantly improve PEB by utilizing a variety of creative processes, including GSL, EP, and CFGC. Moreover, businesses also need to link suitable leadership styles and strengthen the PEB of employees.

*5.2. Practical Implications*

This study offers guidelines for establishing PEB in SMEs. First, promoting green practices requires GSL. To successfully implement eco-initiatives, PEB, and green interests, SME administration needs to cultivate an environmentally oriented leadership mindset among their managers and overall management. Followers genuinely embrace their leader's environmental concern, and need to be motivated by their own passion for the cause. This inner drive will compel them to exhibit PEB, both in the presence of their leader and even when they are distant from them [25]. At the organizational level, firms can greatly benefit from the study's findings; multiple training programmes and effective planning procedures can achieve this milestone. Moreover, the adoption of GSL should link PEB both within and outside the workplace. Organizations are encouraged to provide training to their managers to learn the principles of GSL, enabling them to actively promote PEB [81]. The administration of SMEs should also give special consideration to pro-environmental issues in its hiring and recruitment practices for managers and leaders [28]. To ensure the success of GSL, appropriate reward systems must also be put in place [32]. Second, to encourage pro-environmental actions such as EP and CFGC for the environment, SME management must foster green values among followers through mentoring and training.

Third, this study emphasizes the significance of creating and maintaining CFGC to promote environmentally friendly behaviour [22].

This study draws attention to the important point that an organizational culture that values and provides CFGC creates ideal circumstances for increasing green behaviours [31,60]. The role of GSL is so important, and such leaders need to impart green empowerment and green support to employees so that this can improve green results inside the organization. It is crucial that SME firms try to communicate their green goals and values via explicit organizational rules, such as the construction of the CFGC. SME management should therefore effectively convey these interests and concerns to followers; organizations should incorporate these environmental issues and concerns into their organizational vision, goal, and reward systems.

### 5.3. Limitations and Future Research Directions

This study has several limitations. First, this study was conducted within the context of Pakistan, a developing country, and data were collected from one province; therefore, it is recommended for future research studies to consider the other provinces in Pakistan to validate and provide generalizations. Moreover, the same conceptual framework can be tested in other industries such as education, the banking sector, and the health sector. This study has not tested the impact of heterogeneity among the different types of SMEs. Future research could also investigate the impact of heterogeneity among the different types of SMEs and its analysis. Last but not least, cultural considerations were not considered while examining the connection between GSL and PEB. Future research may consider the cultural component. In order to further grasp the complexity of these variables, future research may perform longitudinal studies, since this study was cross-sectional in nature.

**Author Contributions:** Conceptualization, S.H.A.S. and M.F.; methodology, S.H.A.S. and E.Z.R.; software, M.A. (Muhammad Akram), K.J. and E.Z.R.; validation, M.A. (Mohammed Aljuaid) and M.A. (Muhammad Akram); formal analysis, M.F.; investigation, S.H.A.S.; resources, E.Z.R.; data curation, M.A. (Muhammad Akram); writing—original draft preparation, S.H.A.S. and M.F.; writing—review and editing, M.F. and M.A. (Mohammed Aljuaid); visualization, J.A. and E.Z.R.; supervision, M.F. and J.A.; project administration, S.H.A.S. and M.F.; funding acquisition, M.A. (Mohammed Aljuaid). All authors have read and agreed to the published version of the manuscript.

**Funding:** We would like to extend our appreciation to King Saud University for funding this work through the Researcher Supporting Project (RSP2023R481), King Saud University, Riyadh, Saudi Arabia.

**Institutional Review Board Statement:** The study complied with the Declaration of Helsinki and followed its ethical codes for individuals, samples, and data collection involved in each research procedure. Before the initiation of this study, we presented the study topic to the Ethics Committee of the Bahria University and submitted a proposal stating the purpose of the study, sample, data sources, and details of written informed consent for respondents. All of the above documents were approved by this committee.

**Informed Consent Statement:** Prior to the questionnaire, the researchers asked the respondents to read the written informed consent carefully, introduced the purpose of the study to the respondents, and explained that the data would be used for research only and that all information about the respondents would be kept confidential. All respondents were informed and volunteered to complete the questionnaire.

**Data Availability Statement:** The original contributions presented in the study are included in the article; further inquiries can be directed to the corresponding author.

**Conflicts of Interest:** The authors declare that the research was conducted in the absence of any commercial or financial relationships that could be construed as a potential conflict of interest.

## Abbreviations

| | |
|---|---|
| SMEs | Small and Medium Enterprises |
| GDP | Gross Domestic Product |
| GSL | Green Servant Leadership |
| PEB | Pro-Environmental Behaviour |
| EP | Environmental Passion |
| CFGC | Climate for Green Creativity |
| SLT | Social Learning Theory |

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
