# Peer review of "Impact of Green Servant Leadership in Pakistani Small and Medium Enterprises: Bridging Pro-Environmental Behaviour through Environmental Passion and Climate for Green Creativity"

_sustainability, doi:10.3390/su152014747_

Round 1

Reviewer 1 Report (Previous Reviewer 2)

1. The study addressed Impact is not indicated in the Title.

2. The study objectives are not clear in both the Abstract and Introduction last paragraphs 90 -97 These three contributions seem like study objectives. Usually, the 1st paragraph has to indicate the study objectives apparently. 

3. In-text citations are not in the References. Take care of them.

On Page 2 - Shah et al., 2020, Jamshid et al., 2022, Luu, 2020, Vallernand et al., 2003, in the References Vallernand et al. 2007. Cross check.

On Page 3 - Stern 2000, Faraz et al., 2020, Van Dierendonck, et al., 2017, Luu, 2020, Robert and Barling, 2017

On Page 4 - Stern 2000, Li et al., 2019, Bandura & Walters, 1977, Luu, 2019, Walumbwa et al., 2008, Judge & Piccolo, 2004, Pradhan et al., 2017 but in the References Basu E,......Check it again.

On Page 5 - Yin et al., 2021, Khan et al., 2021, Luu, 2019

On Page 6 - SMEDA, 2022

On Page 7 - Liden, Wayne, Zhao, and Henderson, 2008

On Page 10 - Faraz et al.,2020, Ghazali et al., 2022, Shah et al., 2020

On Page 11 - Yin, 2021, 

Delete Totally

These two Citations are not hanging alone in the References only. Delete totally from the References.

Bissing-Olson, ......2013.

SMEDA, 2017

Paraphrase the following Sentences indicating the Left side Numbers

Abstract: 15-16 The role of SME.....Pakistan GDP. 

                  24 - 27 This study findings.....behaviour of employees.

On Page 3: 113 - 115 Similarly, it is further..... carry out voluntarily.

On Page 5: 224 - 225: For instance,........as CFGC.

On Page 9: 366 - 368: This study ............supporting H1. 372 - 373: Our findings..... H2 and H3. 

Figure 1: Conceptual Framework Dependent Variable Green Servant Leadership mus be at the centre and the rest of the 3 Independent variables were in the boxes with straight and Dotted lines. The impact is not addressed in the CFW. Revisit the CFW again.

Table 2:Questionnaires distributed 850 and received 460 from 6 Clusters. What is the sample formula was used to select the sample size/

3.2. Data Collection Instruments

"I" personal pronoun has to be avoided in the research manuscripts. The author can say that the Researcher take stairs... No. 287

 Table 7: Pathy Analysis and Table 8 Mediation effects that all the Hypotheses are highly significant 

The conclusion has to be one separate paragraph after 5.3. Limitations because all these two pages of the paragraphs have explained the results and discussion only. 

Take care of spelling mistakes here and there. The Citation Author Name Absoramdan or Abramdan?

List of Abbreviations

CFGC, SMEDA, SPSS AMOS can be included in the list of abbreviations.

Take care of the spelling of the Authors and sentences etc.

Author Response

Reviewer 1.

Paper Title: Green Servant Leadership in Pakistani SMEs: Bridging Pro-Environmental Behavior Through Environmental Passion and Creative Climate

Note for the reviewer: We have changed the title of the paper because of one of the reviewer recommended to insert the word of impact in title so the new title of the paper is

“Impact of Green Servant Leadership in Pakistani SMEs: Bridging Pro-Environmental Behavior Through Environmental Passion and Creative Impact Climate”

Comment 1: The study addressed Impact is not indicated in the Title.

Response: Thank you for your comment. We have revised the title and put the word impact in the title as well.

The new title is “Impact of Green Servant Leadership in Pakistani SMEs: Bridging Pro-Environmental Behavior Through Environmental Passion and Creative Impact Climate”

Comment 2: The study objectives are not clear in both the Abstract and Introduction last paragraphs 90 -97 These three contributions seem like study objectives. Usually, the 1stparagraph has to indicate the study objectives apparently.

Comment 3: In-text citations are not in the References. Take care of them.

Comment: On Page 2 - Shah et al., 2021, Jamshid et al., 2022, Luu, 2020, Vallerand et al., 2003, in the References Vallerand et al. 2007. Cross check.

Response: We have added the missing references from the end references. For your quick review we have placed here as well. Thank you for highlighting it.

Shah, S. H. A., Cheema, S., Al‐Ghazali, B. M., Ali, M., & Rafiq, N. (2021). Perceived corporate social responsibility and pro‐environmental behaviors: The role of organizational identification and coworker pro‐environmental advocacy. Corporate Social Responsibility and Environmental Management, 28(1), 366-377.

Jamshed, K., Shah, S. H. A., Majeed, Z., Al-Ghazali, B. M., & Jamshaid, S. (2022). Role of Green Leadership and Green Training on the Green Process Innovation: Mediation of Green Managerial Innovation. Journal of Xidian University, 16(2), 66-72.

Luu, T. T. (2020). Environmentally-specific servant leadership and green creativity among tourism employees: Dual mediation paths. Journal of Sustainable Tourism, 28(1), 86–109. doi: 10.1080/09669582.2019.1675674

Vallerand, R. J., Blanchard, C., Mageau, G. A., Koestner, R., Ratelle, C., Léonard, M.,

Gagné, M., & Marsolais, J. (2003). Les passions de l’âme: On obsessive and harmonious passion. Journal of Personality and Social Psychology, 85(4), 756–767.

Vallerand, R. J., Salvy, S.-J., Mageau, G. A., Elliot, A. J., Denis, P. L., Grouzet, F. M. E., & Blanchard, C. (2007). On the Role of Passion in Performance. Journal of

Personality, 75(3), 505–534.

Comment: On Page 3 - Stern 2000, Faraz et al., 2020, Yi et al., 2020, Van Dierendonck et al., 2017, Luu, 2020, Robert and Barling, 2017

Response: Thank you for highlighting it. We have included all the references in the manuscript. Thank you for improving our work.

Stern, P. C. (2000). New Environmental Theories: Toward a Coherent Theory of Environmentally Significant Behavior. Journal of Social Issues, 56(3), 407–424.

Faraz et al., 2020 was typo error and this reference has been rectified as Ying et al., 2020 in the whole manuscript

Ying, M., Faraz, N. A., Ahmed, F., & Raza, A. (2020). How does servant leadership foster employees’ voluntary green behavior? A sequential mediation model. International journal of environmental research and public health17(5), 1792

Yi et al., 2020 was typo error and the correct reference is actually Ying et al., 2020

Ying, M., Faraz, N. A., Ahmed, F., & Raza, A. (2020). How does servant leadership Foster Employees' voluntary green behavior? A sequential mediation model. International Journal of Environmental Research and Public Health, 17(5), 1792

Van Dierendonck, D., Sousa, M., Gunnarsdóttir, S., Bobbio, A., Hakanen, J., Pircher Verdorfer, A., ... & Rodriguez-Carvajal, R. (2017). The cross-cultural invariance of the servant leadership survey: A comparative study across eight countries. Administrative Sciences7(2), 8.

Luu, T. T. (2020). Environmentally-specific servant leadership and green creativity among tourism

employees: Dual mediation paths. Journal of Sustainable Tourism, 28(1), 86–109.

Robertson, J. L., & Barling, J. (2017). Contrasting the nature and effects of environmentally specific and general transformational leadership. Leadership & Organization Development Journal, 38(1), 22–41.

Comment: On Page 4 - Stern 2000, Li et al., 2019, Bandura & Walters,1977, Luu, 2019, Walumbwa et al., 2008, Judge & Piccolo,2004, Pradhan et al., 2017 but in the References BasuE,......Check it again.

Response: Response: Thank you for highlighting it. We have included all the references in the manuscript. Thank you for improving our work.

Moreover, we checked and Pradhan et al., 2017 is separate reference and we have included in the manuscript.

Stern, P. C. (2000). New Environmental Theories: Toward a Coherent Theory of Environmentally Significant Behavior. Journal of Social Issues, 56(3), 407–424.

Li, D., Zhao, L., Ma, S., Shao, S., & Zhang, L. (2019). What influences an individual’s pro-environmental behavior? A literature review. Resources, Conservation and Recycling, 146, 28-34.s

Bandura, W. (1977). Bandura, A., Walters, RH (1977). Social learning theory (Vol. 1).

Luu, T. T. (2019b). Effects of environmentally-specific servant leadership on green performance via green climate and green crafting. Asia Pacific Journal of Management. 1-29.

Walumbwa, F. O., Avolio, B. J., & Zhu, W. (2008). How transformational leadership weaves its influence on individual job performance: The role of identification and efficacy beliefs. Personnel psychology61(4), 793-825.

Judge, T. A., & Piccolo, R. F. (2004). Transformational and transactional leadership: a meta-analytic test of their relative validity. Journal of applied psychology89(5), 755.

Pradhan, R. K., Panda, P., & Jena, L. K. (2017). Purpose, passion, and performance at the workplace: Exploring the nature, structure, and relationship. The Psychologist-Manager Journal20(4), 222.

Comment: On Page 5 - Yin et al., 2021, Khan et al., 2021, Luu, 2019

Response: Response: Thank you for highlighting it. We have included all the references in the manuscript. Thank you for improving our work

Yin et al., 2021 was a typo error and the correct reference is the Faraz et al., 2021 and it has been rectified as:

Faraz, N. A., Ahmed, F., Ying, M., & Mehmood, S. A. (2021). The interplay of green servant leadership, self‐efficacy, and intrinsic motivation in predicting employees’ pro‐environmental behavior. Corporate Social Responsibility and Environmental Management, 28(4), 1171-1184.

Khan, M. M., Ahmed, S. S., & Khan, E. (2021). Green spillover: Deriving pro-environmental behavior on job and off-job through environmental specific servant leadership. Pakistan Business Review23(1), 1-26.

Luu, T. T. (2019). Building employees’ organizational citizenship behavior for the environment: The role of environmentally-specific servant leadership and a moderated mediation mechanism. International Journal of Contemporary Hospitality Management31(1), 406-426.

Comment:  On Page 6 - SMEDA, 2022

Small and Medium Enterprises Development Authority. (2022). SME sector genesis, challenges and prospects. Islamabad, Pakistan: SMEDA. Available at: http://www.smeda.org/index.dic?option=com_download&view=category&educ=28&Itmid=1052 (Accessed 4th May, 2022)

Comment:  On Page 7 - Liden, Wayne, Zhao, and Henderson, 2008

Liden, R. C., Wayne, S. J., Zhao, H., & Henderson, D. (2008). Servant leadership: Development of a multidimensional measure and multi-level assessment. The leadership quarterly19(2), 161-177.

Comment: On Page 10 - Faraz et al.,2020, Ghazali et al., 2022, Shah et al., 2020

Faraz et al., 2020 was typo error and this reference has been rectified as Ying et al., 2020 in the whole manuscript.

Ying, M., Faraz, N. A., Ahmed, F., & Raza, A. (2020). How does servant leadership foster employees’ voluntary green behavior? A sequential mediation model. International journal of environmental research and public health17(5), 1792

Ghazali et al., 2022, was typo error and this reference has been rectified as Al-Ghazali et al., 2022)

Al-Ghazali, B. M., Gelaidan, H. M., Shah, S. H. A., & Amjad, R. (2022). Green transformational leadership and green creativity? The mediating role of green thinking and green organizational identity in SMEs. Frontiers in Psychology13.

Shah et al., 2020 was typo error and this reference has been rectified as Shah et al., 2021.

Comment: On Page 11 - Yin, 2021,

Response: Thank you for pointing it out. We have rectified it.

Yin et al., 2021 was a typo error and the correct reference is the Faraz et al., 2021 and it has been rectified as:

Faraz, N. A., Ahmed, F., Ying, M., & Mehmood, S. A. (2021). The interplay of green servant leadership, self‐efficacy, and intrinsic motivation in predicting employees’ pro‐environmental behavior. Corporate Social Responsibility and Environmental Management, 28(4), 1171-1184.

Ying, M., Faraz, N. A., Ahmed, F., & Raza, A. (2020). How does servant leadership foster employees’ voluntary green behavior? A sequential mediation model. International journal of environmental research and public health17(5), 1792.

Comment: These two Citations are not hanging alone in the References only. Delete totally from the References.

Bissing-Olson, ......2013.

Response: Thank you for pointing it out. We have included in this reference in the below sestion of manuscript. Thank you.

  1. Theoretical background and hypotheses development 2.1. Pro-environmental behaviour PEBs are generally regarded as to be human behaviours that are developed and used sustainably in relation to the environment or that attempt to lessen the negative effects of those behaviours on the environment (Li et al., 2020; Bissing-Olson et al., 2013).

Response: Thank you for pointing it out. We have removed this reference of SMEDA, 2017.

Comment: Paraphrase the following Sentences indicating the Left side Numbers

Comment:  Abstract: 15-16 The role of SME.....Pakistan GDP.

24 - 27 This study findings.....behaviour ofemployees

Response: Abstract: Small and Medium Enterprises (SMEs) are crucial for any economy to grow and succeed, as evidenced in the Pakistani context where SMEs contribute 30% to the country's GDP. Objective of the study is to link the green servant leadership to pro-environmental behaviour of employees, particularly in SMEs which very few study have investigated. Building on the Social Learning Theory, this study developed and tested the conceptual framework that examine the impact of green servant leadership (GSL) on pro-environmental behaviour (PEB) and the mediating role of environmental passion and climate for green creativity among employees of Small and Medium-sized Enterprises (SMEs). Data were collected from 460 middle line managers, and structural equation modelling technique was applied to test hypotheses. Main findings revealed that green servant leadership impacted by the PEB, while environmental passion and climate for green creativity mediated these relations. The study findings demonstrated that a GSL with strong practices and values towards environment could have a significant impact on employees’ PEB. This study filled research gaps in different ways. First by identifying the role of green servant leadership on pro-environmental behaviour of employees. Second, by examining the dual mediation mechanism of environmental passion and climate for green creativity between the green servant leadership and pro-environmental behaviour. Third, this study focused an economic context of a developing country. The study offers guidelines for establishing pro-environmental behaviours in the SMEs. Multiple training programmes and effective planning procedures can achieve this milestone. The administration of SMEs should also give special consideration to pro-environmental issues in its hiring and recruitment practices for managers and leaders.

Comment:  On Page 3: 113 - 115 Similarly, it is further..... carry out voluntarily.

Response:

Response: Thank you for pointing it out. We have change it to this. Thank you.

Additionally, it is posited that the conduct observed within the organization, which deviates from formal policies and prescribed implementation procedures, is undertaken voluntarily by employees (Kim et al., 2017).

Comment: On Page 5: 224 - 225: For instance,........as CFGC.

Response: Thank you for pointing it out. We have change it to this. Thank you.

Moreover, the allocation of financial resources by the organization for green projects and initiatives are referred as CFGC. It has been argued that the values of a green environment convey to employees the need for them to act sustainably (Norton et al., 2017).

Comment: On Page 9: 366 - 368: This study ............supporting H1. 372 -373: Our findings..... H2 and H3.

Response: Thank you for highlighting this important point. We have removed the “Our” word and addressed this issue throughout the manuscript.

Figure 1: Conceptual Framework Dependent Variable Green Servant Leadership must be at the centre and the rest of the3 Independent variables were in the boxes with straight and Dotted lines. The impact is not addressed in the CFW. Revisit the CFW again.

Response: Thank you and Actually the Green Servant Leadership is the independent variable and environmental passion and Climate for green creativity are mediating variables. While dependent variable is the pro environmental behavior.

The reason the the Green Servant Leadership is the left side and environmental passion and Climate for green creativity are in the middle because of mediating role. The pro environmental behavior is the dependent variable that is the reason it is on the right side. Moreover, Straight lines and dotted lines show the direct effects and indirect effects. Thank you for your comment. We hope that we explained well.

Table 2:Questionnaires distributed 850 and received 460from 6 Clusters. What is the sample formula was used to select the sample size/

Response: Thank you for your comment..

For sampling, actually 360 enterprises were chosen because they agreed to be part of this research study. Moreover, out of these 6 industries and among 430 enterprises the middle line managers are more than 50,000 and sample size according to the Morgan and Krejcie sample size table, (1970), the sample size should be above the 381 for more than 50,000 employees. So that is why we wanted to have greater sample size than 381. Moreover, the reason to distribute the 850questionnaire is because of the low questionnaire return rate, we wanted to have better response rate and for that reason we distribute more than required number of questionnaires, just to avoid the low number of response rate.

3.2. Data Collection Instruments

"I" personal pronoun has to be avoided in the research manuscripts. The author can say that the Researcher take stairs... No. 287

Response: Thank you for your comment. Actually, those are the statements (items) of the questionnaire that we have adapted from different studies and we used to asked the respondents to answer, so these questions (Items) are part of the questionnaire.

Table 7: Pathy Analysis and Table 8 Mediation effects that all the Hypotheses are highly significant

Response: Thank you for your comment and appreciating our work.

The conclusion has to be one separate paragraph after 5.3.Limitations because all these two pages of the paragraphs have explained the results and discussion only.

Response: Thank you for the comment. Actullay we followed the journal template and in which limitation is the last part. Moreover, we have discussed and concluded based on the study findings as well and now after having receiving 5/ five reviewers comment. This discussion and conclusion part has already improved as well. It is given for your quick review. Thank you for your comment respected reviewer.

Comment: Take care of spelling mistakes here and there. The Citation Author Name Absoramdan or Abramdan?

Response: Thank you for highlighting this point. We have already proofread the complete manuscript. Moreover, we have also rectified the citation Aboramadan et al., 2021 as well.

Note: We would like to thank the respected reviewer for his/her valuable comments because his/her recommendations our study has improved a lot. Thank you once again.

Reviewer 2 Report (Previous Reviewer 3)

This manuscript has been improved. I still recommend clearly stating the line or page number of the correction.

Author Response

Reviewer 2.

Paper Title: Green Servant Leadership in Pakistani SMEs: Bridging Pro-Environmental Behavior Through Environmental Passion and Creative Climate

Note for the reviewer: We have changed the title of the paper because of one of the reviewer recommended to insert the word of impact in title so the new title of the paper is

“Impact of Green Servant Leadership in Pakistani SMEs: Bridging Pro-Environmental Behavior Through Environmental Passion and Creative Impact Climate”

Comments and Suggestions for Authors: This manuscript has been improved. I still recommend clearlystating the line or page number of the correction.

Response : Thank you for your comment and appreciating our work. We have now rectified the page number.

Reviewer 3 Report (New Reviewer)

It is quality paper because it focused an economic context of not only Pakistan but also other developing country. In additional, this study has offered guidelines for establishing pro-environmental behaviours in the SMEs in the era of Green economy.

Author Response

Reviewer 3.

Paper title: Green Servant Leadership in Pakistani SMEs: Bridging Pro-Environmental Behavior Through Environmental Passion and Creative Climate

Note for the reviewer: We have changed the title of the paper because of one of the reviewer recommended to insert the word of impact in title so the new title of the paper is

“Impact of Green Servant Leadership in Pakistani SMEs: Bridging Pro-Environmental Behavior Through Environmental Passion and Creative Impact Climate”

Comments and Suggestions for Authors: It is quality paper because it focused an economic context of not only Pakistan but also other developing country. In additional, this study has offered guidelines for establishing pro-environmental behaviours in the SMEs in the era of Green economy.

Response: Thank you for your comment and appreciating our work. We have now rectified the page number.

Note: We would like to thank the respected reviewer for his/her valuable comments because his/her recommendations our study has improved a lot. Thank you once again.

Reviewer 4 Report (New Reviewer)

Please read comments in pdf file

Author Response

Paper title: Green Servant Leadership in Pakistani SMEs: Bridging Pro-Environmental Behavior Through Environmental Passion and Creative Climate

Note for the reviewer: We have changed the title of the paper because of one of the reviewer recommended to insert the word of impact in title so the new title of the paper is

“Impact of Green Servant Leadership in Pakistani SMEs: Bridging Pro-Environmental Behavior Through Environmental Passion and Creative Impact Climate”

Reviewer 4

Comment 1: Shouldn´t it be GSL?

Response 1: Thank you for your valuable comment. We have revised this to GSL.

Comment 2: A t is left over

Response 2: Thank you for highlighting it. We have removed it and further we proofread the whole paper.

Comment 3: try not to repeat same word of the following sentence

Response: Thank you for highlighting it. We have revised it.

Comment 4: Maybe it should be managers

What kind of employees answered the questions?, How old were there, what are they role in the company?, how long have they worked there?, what level of studies do they have?, What is the percentage of males and females?

Response 4: Thank you for your valuable comment. Unit of analysis of this study is the managerial level employees (middle managers’), we carefully selected only those managerial level employees’, who are working with organization for the past two years atleast.

We have also added the new table in which we have described the whole details of the demographics of the study. The table is also given below for your quick review.

Table 4: Demographics

Demographics

No. of Respondents

Percentage (%)

Gender:

Male

326

71

Female

134

29

Age:

Less than 40 years

312

68

More than 40 years

148

32

Education:

Bachelors

386

84

Masters.

74

16

Experience:

< 5 years

165

36

5-10 years

203

43

  >10 years                                

97                            

21           

Comment: 5. Try to be more precise, position and on what statistic, maybe these should go on the introduction

What is the situation now?, what amount is being invested?

You need to show values instead of vague adjectives

(Discussion and Conclusion

This study advances the discussion on PEB, especially in the context of energy, by taking into account Pakistan's environmental fragility and the role that citizens play in maintaining the environment and ecosystem. Together with other nations from the Global South, Pakistan is listed lower on the list of nations with better sustainability conditions. For a more favourable and environmentally friendly future, the country requires support from all economic sectors with its sustainability activities.)

Response 5: Thank you for highlighting this issue. We have deleted those statements which were highlighted by respected reviewer.  

Comment 6: Please revise this sentence

since this study has been conducted in Pakistan which is developing country context while data has been collected from one province”

Response 6: Thank you for highlighting this issue. We have revised the sentences and it is also given below for your quick review. Thank you.

5.3. Limitations and future research directions

This study has several limitations. First, First, this study was conducted within the context of Pakistan, a developing country and data were collected from one province, therefore, therefore it is recommended for future research studies to consider the.

Note: We would like to thank the respected reviewer for his/her valuable comments because his/her recommendations our study has improved a lot. Thank you once again.

Reviewer 5 Report (New Reviewer)

This paper wants a try to use the data from 460 managers in Pakistani SMEs to prove that environmental passion and creative climate is good for improve pro-environment behavior in green servant behavior, the results sound reasonable. However, authors need provide more information to support this study: SEM figures in the middle and end, observed variables and questions in sample statement, environment data and situation analysis of Pakistan in the introduction, brief and valuable suggestions in the abstract, Analysis of heterogeneity among the different types of SMEs. 

Author Response

Paper title:

Green Servant Leadership in Pakistani SMEs: Bridging Pro-Environmental Behavior Through Environmental Passion and Creative Climate

Note for the reviewer: We have changed the title of the paper because of one of the reviewer recommended to insert the word of impact in title so the new title of the paper is

“Impact of Green Servant Leadership in Pakistani SMEs: Bridging Pro-Environmental Behavior Through Environmental Passion and Creative Impact Climate”

Reviewer 5

Comments and Suggestions for Authors

This paper wants a try to use the data from 460 managers in Pakistani SMEs to prove that environmental passion and creative climate is good for improve pro-environment behavior in green servant behavior, the results sound reasonable. However, authors need provide more information to support this study: SEM figures in the middle and end, observed variables and questions in sample statement, environment data and situation analysis of Pakistan in the introduction, brief and valuables suggestions in the abstract, Analysis of heterogeneity among the different types of SMEs.

Comment: However, authors need provide more information to support this study:

Response 1: Thank you for your comment. We have added the more information in introduction section and in the discussion section to support this study.

Moreover, we have included new references in introduction, discussion and implication part to further strengthen the study. The new references are given below for your quick review. Thank you for improving our work.

References

  1. Tu, Y., Li, Y., & Zuo, W. (2023). Arousing employee pro‐environmental behavior: A synergy effect of environmentally specific transformational leadership and green human resource management. Human Resource Management62(2), 159-179.
  2. Alyahya, M., Aliedan, M., Agag, G., & Abdelmoety, Z. H. (2023). The antecedents of hotels’ green creativity: the role of green HRM, environmentally specific servant leadership, and psychological green climate. Sustainability15(3), 2629.
  3. Thabet, W. M., Badar, K., Aboramadan, M., & Abualigah, A. (2023). Does green inclusive leadership promote hospitality employees’ pro-environmental behaviors? The mediating role of climate for green initiative. The Service Industries Journal43(1-2), 43-63.
  4. Afridi, A., Shahjehan, A., Zaheer, S., Khan, W., & Gohar, A. (2023). Bridging Generative Leadership and Green Creativity: Unpacking the Role of Psychological Green Climate and Green Commitment in the Hospitality Industry. SAGE Open13(3), 21582440231185759.
  5. Elzek, Y. S., Soliman, M., Al Riyami, H., & Scott, N. (2023). Talent management and sustainable performance in travel agents: do green intellectual capital and green servant leadership matter?. Current Issues in Tourism, 1-16.
  6. Khan, A., Hussain, S., & Sampene, A. K. (2023). Investing in green intellectual capital to enhance green corporate image under the Influence of green innovation climate: A Case of Chinese Entrepreneurial SMEs. Journal of Cleaner Production, 418, 138177.

The passage is given below for your quick review.

  1. Introduction

Nowadays organizations are taking the multiple steps to prevent the harmful practices towards natural environment and taking the initiatives to promote the environmentally friendly practices as it has become the centres of the attention for the organization to consider the sustainability and pro-environmental behaviour (Faraz et al., 2021; Ying et al., 2020, Faraz et al., 2021). This type of employee conduct refers to voluntary actions that contribute to the environmental sustainability of the employer's organization (Kim et al., 2017).

Pro-environmental behaviour (PEB) refers to the practices which are helpful in promoting the natural environment through different practices of recycling, reusable initiatives, reprocessing, rebuilding and the applications of different ideas to implement the practices which reduce the harmful effect of organization towards environment by adopting the practices of green products and processes (Faraz et al., 2021; Afsar et al., 2020). Organizations that are involve in the PEB practices are in better position to gain competitive edge over their competitors as the pro environmental practices reduce the costs, generate the revenue, and helps in creating the positive image by certain practices towards the sustainability. Therefore, there is much curiosity among the researchers to investigate further the antecedents and underlying mechanisms that promote PEB (Shah et al., 2020; Faraz et al., 2021; Testa et al., 2020).

The role of leaders is immensely important in shaping the employees behaviour (Robert & Barling, 2013; Afsar et al., 2018; Li et al., 2023). Even though, there is lack of understanding and the empirical evidence to uncover the underlying mechanisms of leadership role on enhancing the PEB (Al-Ghazali et al., 2022; Afsar et al., 2018; Graves & Sarkis, 2018). Particularly the servant leadership (Faraz et al., 2021; Ying et al., 2020), which have been reported as significant predictor of PEB (Lee et al., 2020; Hoch et al., 2018). GSL places a higher emphasis on environmental advantages, both for the leader personally and the organization, while simultaneously focusing on instilling pro-environmental values in key organizational stakeholders, such as employees and customers (afsar et al., 2018). Various studies have been conducted on the other style of leadership like green transformational leadership (Li et al., 2020), ethical leadership (Jamshid et al., 2022), responsible leadership (Afsar et al., 2020). Therefore, there is a gap of empirical evidence to investigate the green servant leadership impact on the PEB (Ying et al., 2020; Luu, 2019; Maqbool et al., 2023). The novelty of the study lies in the bridging the gap in literature by focusing on the environmental aspect of the Green Servant Leadership (GSL) that inculcate and promote the PEB.

The role of the passion is significant to carry out different activities with dedication. Employees exhibits the passion in multiple ways like demonstrating the positive emotions at work, developing meaningful connection to different work tasks, and intrinsically directed to accomplish the tasks (Vallerand et al., 2003).  Environmental passion (EP) is the employee’s emotional experiences towards the various environmental practice in the organization (Li et al., 2020). A strong passion for the environment serves as a motivating force that drives individuals to engage in pro-environmental behavior (Khan et al., 2021). Employees who are passionate about environmental causes not only engage in spontaneous pro-environmental actions but also maintain a consistent commitment to PEB, exhibits themselves as environmentalists (Afsar et al., 2016). Employees with strong EP not only inclined towards the PEB but also identify themselves as environmentalists (Li et al., 2020; Afsar et al., 2016). The role of leader is pertinent to further strengthen the EP in employees which in turn can promote the PEB. GSL specifically prioritizes environmental concerns and acknowledges employees' contributions to the community, particularly through environmentally conscious actions, can enhance their EP as conscientious environmental citizens and portray PEB in the organization (Afsar et al., 2018). There is lack of the studies and gap which investigate the mediation mechanism of EP between the GSL and PRB particularly in the Small and Medium Enterprises (SMEs) of Pakistan.

To promote the PEB the role of Climate for Green Creativity (CFGC) is also crucial (Aboramadan et al., 2021; Choong et al., 2019). CFGC is refer to the organizational support to employees to achieve their outcomes through their own creative manner (Aboramadan et al., 2021; Datu, & Buenconsejo, 2021; Luu, 2019b). Particularly, when employees believe that they that are appreciated and rewarded at workplace for their creativity, such practices enhance the notion of the climate for creativity (Kim & Yoon, 2015; Luu, 2019a). GSL are regarded as architects of environments that promote a climate conducive to creativity (Karatepe et al., 2020). Moreover, they provide the essential resources and offer support for creative and innovative endeavours (Aboramadan et al., 2021). Servant leadership was found to have positive relationship with group creativity (Linuesa-Langreo et al., 2016). In an environment where there is an abundance of resources, comprehensive support, and attractive incentives for novel ideas, employees are more inclined to engage in innovative behaviors while considering the environment (Aboramadan et al., 2021)+. However, there is a limitation in the literature regarding the mechanism underlying the mediation of CFGC between the GSL and PEB (Aboramadan et al., 2021; Luu, 2020).

This study has several contributions and objectives of this study: First, this to investigate the GSL impact on the PEB which are rarely investigated in the context of Pakistan which is developing country. Second drawing on the theory of social learning theory, this study examines the underlying mechanism of EP between GSL and PEB and integrates the holistic research work in one framework which is another contribution of this study. Third contribution of the study lies in the investigation of the mediating mechanism of the climate for the green creativity between the GSL and PEB in the Pakistan context which is the developing country.

  1. Discussion and Conclusion

This study advances the discussion on PEB, especially in the context of energy, by taking into account Pakistan's environmental fragility and the role that citizens play in maintaining the environment and ecosystem. The findings demonstrated that GSL can have an impact on employees' PEB inside a firm.

Different organizational components, such as leadership style and a welcoming green workplace, might influence the energy consumption behaviour of SMEs workers in this industry (Al-Ghazali et al., 2022). This means that leaders who put sustainability first and encourage and help their employees do things that are good for the environment can have a big impact on how employees in the SME sector act. The fact that SMEs use a lot of energy to make things and provide services suggests that these businesses may have a big effect on the environment. Because of this, it's important to think about how organizational factors like leadership style and a friendly, green workplace affect the way SME workers in this industry use energy. A leadership style that emphasizes sustainability and servant leadership can encourage employees to do things that are good for the environment, like using less energy, recycling, and making less trash.

This study findings demonstrated that a GSL with strong practices and values towards environment could have a significant impact on employees' PEB (β= 0.687) supporting H1. GSL can be defined as a leader who values environmental advantages over financial ones. They also demonstrate their dedication to the environment by their energy consumption tendencies. Through the lens of social learning process theory, employees involve in energy conservation activities when they observe that their leader care and promote PEB, which ultimately motivates them to exhibits PEB into practice. The findings indicated that GSL positively impact on EP and CFGC, supporting H2 and H3. This suggests that GSL incite EP and CFGC which further encourages individuals to develop favourable perceptions for the environment and exhibit PEB (Absoramadan et al., 2021). Another interesting finding is the CFGC found to have positive and favourable impact on PEB, supporting H4 and EP found to have also positive impact on PEB, supporting H4 and H5, respectively. The findings are consistent with the social learning process, which holds that followers mimic leaders by adopting their behavior and actions, and that this leads to employees adopting green behavior, which prompts them to participate in PEB (Aboramadan et al., 2021). The statistical findings in this regard demonstrated that EP not only significantly mediates the association between GSL and PEB (β = 0.541), but also directly effects PEB (b = 0.58), supporting H6. Similarly, the findings show the mediation effect of CFGC (β = 0.483), between GSL and PEB (β = 0.483), and effects directly PEB (β = 0.57), supporting H7. This implies that the influence of EGSL on fostering PEBis intricate, with a mediating mechanism playing a pivotal role in this process (Tu et al., 2023). This led to the notion that GSL not only guides and facilitates their employees to reduce their energy consumption but also enhance employees' perceptions of how their energy use can help their firm to have an impact on their PEB. GSL emerges as a valuable leadership approach, as it nurtures environmental enthusiasm among employees, subsequently leading them to exhibit pro-environmental behaviors (Khan et al., 2021). GSL incites environmental passion about the environment, and as a result, such employees exhibit pro-environmental behaviour (Khan et al., 2021; Li et al., 2020).

Environmental passion refers to employees' emotional attachment and commitment to environmental sustainability. "Culture for green creativity" refers to the organizational culture that encourages and supports innovation and creativity in developing environmentally sustainable practices. These factors can mediate the effect of GSL on PEB by influencing the motivation and ability of employees to engage in environmentally sustainable practices. GSL may have a significant impact on PEB, and this relationship may be facilitated by factors such as EP and a CFGC. It is evident that a green creativity-friendly environment serves as a mechanism for conveying expectations to employees regarding pro-environmental behaviors, such as organizational citizenship behavior for the environment (Alyahya et al., 2023; Thabet et al., 2023). By fostering a culture of environmental sustainability and providing employees with the necessary resources and support, SMEs in Pakistan can promote PEB among their employees, which can have significant environmental and social benefits. The purpose of the proposed model was to investigate the effects of GSL on PEB directly and through mediating role of EP and CFGC. Findings of the study is consistent with the previous studies (Absoramadan et al., 2021; Ying et al., 2020; Karatepe et al., 2020). GSL offers employees abundant and valuable green resources, consequently stimulating their perception of a climate conducive to creativity. This, in turn, motivates employees to participate in PEB (Afridi et al., 2023). Previous studies have focused on organizational practices to promote the PEB, but this study focused on the role of GSL on overall PEB.

EP is important for SMEs in Pakistan because it can be a strong motivator for employees to act in ways that are good for the environment. When employees have a strong emotional connection to and commitment to protecting the environment, they are more likely to take the initiative and go above and beyond what is expected of them at work to do things that are good for the environment. When it comes to SMEs in Pakistan, where there may be a lack of resources and skills for putting environmentally sustainable practices into place, passion for the environment can be a big driver of change. By encouraging employees to care about the environment, SMEs can create a culture where sustainability is valued and put first. This can make people more motivated and committed to protecting the environment, which can lead to less energy use, less waste, and other actions that are good for the environment. Also, in a country like Pakistan, where environmental problems like air and water pollution, deforestation, and climate change are big, promoting EP can help raise awareness and motivate employees, customers, and the wider community to take action. In the end, encouraging SME employees in Pakistan to care about the environment can help build a more sustainable and resilient future. 

Furthermore, the CFGC is important for SMEs in Pakistan because it can foster an organizational culture that encourages and supports innovation and creativity in developing environmentally sustainable practices. Promoting a CFGC can also help SMEs in Pakistan overcome barriers such as a lack of knowledge, experience, and resources. By encouraging employees to develop and test new ideas and solutions, SMEs can tap into the collective knowledge and creativity of their workforce to find practical and effective ways to reduce their environmental impact

Comment: questions in sample statement

Response 2: Moreover, the sample statements have been added in the methodology section. The passage is given below for your quick review.

3.2. Data collection instruments

To assess GSL measured though a twelve-item scale adopted from study of Liden, Wayne, Zhao, and Henderson, (2008). The sample statements are “My manager emphasizes the importance of contributing to the environmental improvement” and “I am encouraged by my manager to volunteer in the environmental activities”. Items for the pro-environmental behaviour scale were taken from Robertson and Barling, (2013). There was a total of 12 items used to assess pro-environmental behaviour. Some examples are: “At work, I take stairs instead of elevators to save energy". Moreover, 10-item scale established by Roberson and Barling, (2013) was used to assess EP.  Sample items include “I am passionate about the environment,” In order to measure the CFGC, this study adopted the 5item scale from Kim and Yoon, (2015). The sample statement is "I am passionate about the environment."

Response 3: In addition to that, we have added information in introduction section, implication section as well.

The passage is given below for your quick review.

5.1. Theoretical implications

This study added several new insights to the body of knowledge about environmental management in SMEs settings. First, this study filled the critical knowledge gap and addressed the scholarly call made by Ghazali et al., (2022) to further explore how GSL may encourage pro-environmental behaviour in the SEMs sector. As findings of this study provided empirical evidences that GSL is capable of effecting the desired change. This study adds to the existing body of knowledge. Furthermore, employees are positively influenced by the environmental concern displayed by their leader through GSL (Elzek et al., 2023). Second, there is a lot of attention being paid to the research on the antecedents of PEB which is still considered as it is in early stages and needs further investigations (Ghazali et al., 2022; Shah et al., 2020). Therefore, this study highlights the antecedents of pro-environmental behaviours, namely EP and CFGC for the PEB in SMEs.

Moreover, the study revealed that GSL instigates EP, which, in turn, motivates employees to adopt PEB. the leader's genuine environmental care influences the followers' passion for the environment (Li et al., 2020; Khassawneh & Elrehail, 2022). Third, although research in the SMEs has shown that GSL is a key factor in employees' pro-environmental activities, however, little is known about the underlying mechanisms that control these associations (Mughal et al., 2022). Furthermore, it was found that the GSL is a leadership approach that can effectuate such desired transformation. Employees are affected by the environmental awareness demonstrated by their leader through GSL (Afsar et al., 2018). Fourth, this study adds to the social learning theory in the SMEs because there is little evidence regarding how employees EP and CFGC lead to PEB and their conceptualization. As servant leaders provide green-related resources, these resources are disseminated among employees, ultimately fostering positive perceptions among them regarding the environment's creative climate. When environmentally-focused GSL offer ample environmental resources, it can act as a catalyst for resource exchange among employees. This, in turn, contributes to the development of a shared mental model and ultimately enhances a positive perception of the organizational climate (Khan et al., 2023). Fifth, no current study has attempted into the influence of EP in mediating the relationship between GSL and PEB in context of SMEs particularly (Khan et al., 2021; Li et al., 2020; Datu, & Buenconsejo, 2021; Faraz et al., 2021; Choong et al., 2019). According to the findings of this study, SMEs may significantly improve PEB by utilizing a variety of creative processes including GSL, EP and CFGC. Moreover, businesses also need to link the suitable leadership styles and strengthen the PEB of employees.

5.2. Practical implications

This study offers guidelines for establishing pro-environmental behaviours in the SMEs. First, promoting green practices requires GSL. To successfully implement eco-initiatives, PEB and green interests, SMEs administration needs to cultivate an environmentally oriented leadership mindset among their managers and overall management. Followers genuinely embrace their leader's environmental concern and needs to be motivated by their own passion for the cause. This inner drive will compel them to exhibit PEB, both in the presence of their leader and even when they are distant from them (Khan et al., 2021. At the organizational level, firms can greatly benefit from the study's findings, multiple training programmes and effective planning procedures can achieve this milestone. Moreover, adoption of GSL should link pro-environmental behavior both within and outside the workplace. Organizations are encouraged to provide training to their managers to learn the principles of GSL, enabling them to actively promote PEB (Khan et al., 2023). The administration of SMEs should also give special consideration to pro-environmental issues in its hiring and recruitment practices for managers and leaders (Absoramadan et al., 2021). To ensure the success of GSL, appropriate reward systems must also be put in place (Kim & Yoon, 2015). Second, to encourage their pro-environmental actions, such as EP and CFGC for the environment, SMEs management must foster green values among followers through mentoring and training. Third, this study emphasizes the significance of creating and maintaining a climate for green creativity to promote environmentally friendly behaviour (Luu, 2019b).

This study draws attention to the important point that an organizational culture that values and provide climate for green creativity creates ideal circumstances for increasing green behaviours (Luu, 2019a; Chou, 2014). The role of GSL is so important and they need to impart green empowerment and green support to employees so that this can improve green results inside the organization. It is crucial that SMEs firms to try to communicate their green goals and values via explicit organizational rules, as the construction of the climate for green creativity. SMEs management should therefore, to effectively convey these interests and concerns to followers, organizations should incorporate these environmental issues and concerns into their organizational vision, goal, and reward systems.

Response 4: Respected reviewer has pointed rightly to include suggestions in the abstract and we tried to incorporate which we could but due to words limitation (Abstract already has reached to 263 words) we could not add more, however, we have included suggestion in implication section.

Response 5: Regarding your valuable comment of heterogeneity analysis. We did not check the heterogeneity of SMEs because our objectives were to investigate the impact of GSL on PEB directly and through mediating mechanisms, however, we have put it in our limitation and recommended for future research to analyze the heterogeneity among different types of SMEs. Thank you for highlighting these points, our study improved.

The passage is given below for your quick review.

5.3. Limitations and future research directions

This study has several limitations. First, since this study has been conducted in Paki-stan which is developing country context while data has been collected from one province, therefore it is recommended for future research studies to consider the other provinces in Pakistan to validate and provide generalizations. Moreover, the same conceptual frame-work can be tested in other industries such as education, the banking sector, and the health sector. Second, This study has not tested the impact of heterogeneity among the different types of SMEs. Future research could also investigate the impact of heterogeneity among the different types of SMEs and its analysis. Last but not the least, cultural considerations were not considered while examining the connection between GSL and PEB. Future research may con-sider the cultural component. In order to further grasp the complexity of these variables, future research may perform longitudinal studies since this study was cross-sectional in nature.

Note: We would like to thank the respected reviewer for his/her valuable comments because his/her recommendations our study has improved a lot. Thank you once again.

Round 2

Reviewer 4 Report (New Reviewer)

line 168: Shouldn´t it be GSL instead of GLS

line 223: too many times uses the same word employees, apostrophe left over

line 261: categories is joint with Wood, leave a space between words

line 262: Textiles is joint with total, leave a space between words

line 283: leave a space between word and bracket

line 285: leave a space between word and bracket

line 319: leave a space between word and bracket

line 478: there are two First

line 168: Shouldn´t it be GSL instead of GLS

line 223: too many times uses the same word employees, apostrophe left over

line 261: categories is joint with Wood, leave a space between words

line 262: Textiles is joint with total, leave a space between words

line 283: leave a space between word and bracket

line 285: leave a space between word and bracket

line 319: leave a space between word and bracket

line 478: there are two First

Author Response

Thank you for your attentive review.

We have addressed and corrected all the comments you provided, including the errors at lines 168, 223, 261, 262, 283, 285, 319, and 478. We appreciate your commitment to improving the quality of the document. If there are any more suggestions or corrections, please feel free to provide them.

Thank you

Round 3

Reviewer 4 Report (New Reviewer)

Finally it can be published

This manuscript is a resubmission of an earlier submission. The following is a list of the peer review reports and author responses from that submission.

Round 1

Reviewer 1 Report

The authors present their work on sustainibility with various influencing factors, which is an interesting tiopic and can be seen as valid to be implemented into leadership worldwide. In general the manuscript is well written, maybe a few minor errors are there. The use of statistics is appropriate and well conducted.

I only have one comment to improve this work:

1. Please explain the abreviations at first use, e.g. in the abstract you use "SME" already until you finally describe what it is much later.

Reviewer 2 Report

1. Study objectives must be clear in the Abstract.

2. List of Abbreviations is required.

3. Study objectives must be included in the 1st paragraph of the introduction.

4. Paraphrase the indicated sentences.

5. Check the Conceptual Framework and include the Dependent Variable in it.

6. Justify the Sample size argument based on the Structural Equation Model.

7. Include a separate paragraph for the Conclusion.

8. Take care of the editorial and spelling corrections overall manuscript.

9. Citataion corrections in the References.

10. In-text citations have to be included in the References.

Reviewer 3 Report

1. It is recommended to explain why Pakistan SMEs are used as a case. Among them, the study discussed four categories as Wood and Furniture, Sports, Leather and Footwear, and Textiles. Can the results of the generalization and inference to other circumstances?

2. Some works of literature are too old, it is recommended to update them with relevant research in recent years. For example Greenleaf, R. (2007). Martins, E. C., & Terblanche, F. (2003).

3. Table 1. Industry Profile and Table 7. Path Analysis. They are across the page, it is recommended to correct the typesetting.

4. Is there a pre-test of reliability and value for the questionnaire?

5. The questionnaire content of this study adopted is recommended to readers for review.

Reviewer 4 Report

Dear authors,

Thank you for your contribution to the field of GSL as it is an interesting topic. While reading your manuscript, I noticed that the discussion section could benefit from the inclusion of more recent research and a clearer connection to the previous studies you have reviewed. To ensure that your arguments are based on the most up-to-date evidence, I recommend that you incorporate recent studies published in reputable academic journals. In addition, it would be better for readers to separate the discussion section from the conclusion (add a specific section for conclusion at the end). 

 Here are some relevant and direct studies that you might need to use:

1.     Li et al.. (2023). Towards Examining the Link Between Green HRM Practices and Employee Green in-Role Behavior: Spiritual Leadership as a Moderator. Psychology Research and Behavior Management, 383-396.

2.     Maqbool et al., (2022). The role of diverse leadership styles in teaching to sustain academic excellence at secondary level. Frontiers in Psychology13.

3.     Khassawneh, O., & Elrehail, H. (2022). The Effect of Participative Leadership Style on Employees’ Performance: The Contingent Role of Institutional Theory. Administrative Sciences12(4), 195.

Thank you